# Chemical Composition, Antioxidant and Antiproliferative Activities of *Taraxacum officinale* Essential Oil

**DOI:** 10.3390/molecules27196477

**Published:** 2022-10-01

**Authors:** Fatima Zahra Kamal, Radu Lefter, Cosmin-Teodor Mihai, Hanane Farah, Alin Ciobica, Ahmad Ali, Iulian Radu, Ioannis Mavroudis, Abdellah Ech-Chahad

**Affiliations:** 1Laboratory of Physical Chemistry of Processes, Faculty of Sciences and Techniques, Hassan First University, B.P. 539, Settat 26000, Morocco; 2Biomedical Research Center, Romanian Academy, Iași Branch, 8th Carol I Avenue, 700506 Iași, Romania; 3Advanced Centre for Research-Development in Experimental Medicine, Grigore T. Popa University of Medicine and Pharmacy of Iași, 700115 Iași, Romania; 4Department of Biology, Faculty of Biology, Alexandru Ioan Cuza University, 11th Carol I Avenue, 700506 Iași, Romania; 5Department of Life Sciences, University of Mumbai, Vidyanagari, Santacruz (East), Mumbai 400098, India; 6Faculty of Medicine, Grigore T. Popa University of Medicine and Pharmacy, 700115 Iași, Romania; 7Department of Neurology, Leeds Teaching Hospital University, Leeds LS9 7TF, UK

**Keywords:** *Taraxacum officinale*, cervical cancer (HeLa), antiproliferative activity, antioxidant activity, GC-MS

## Abstract

*Taraxacum officinale* (*TO*) has been historically used for medicinal purposes due to its biological activity against specific disorders. To investigate the antioxidant and the antiproliferativepotential of *TO* essential oil in vitro and in vivo, the chemical composition of the essential oil was analyzed by GC-MS. The in vivo antioxidant capacity was assessed on liver and kidney homogenate samples from mice subjected to acetaminophen-induced oxidative stress and treated with *TO* essential oil (600 and 12,000 mg/kg BW) for 14 days. The in vitro scavenging activity was assayed using the 1,1-diphenyl-2-picrylhydrazyl (DPPH) and the reducing power methods. The cytotoxic effects against the HeLa cancer cell line were analyzed. The GC-MS analysis showed the presence of **34** compounds, **8** of which were identified as major constituents. The *TO* essential oil protected mice’s liver and kidneys from acetaminophen-induced oxidative stress by enhancing antioxidant enzymes (catalase, superoxide dismutase, and glutathione) and lowering malondialdehyde levels. In vitro, the *TO* essential oil demonstrated low scavenging activity against DPPH (IC_50_ = 2.00 ± 0.05 mg/mL) and modest reducing power (EC_50_ = 0.963 ± 0.006 mg/mL). The growth of the HeLa cells was also reduced by the *TO* essential oil with an inhibition rate of 83.58% at 95 µg/mL. Current results reveal significant antioxidant and antiproliferative effects in a dose-dependent manner and suggest that *Taraxacum officinale* essential oil could be useful in formulations for cancer therapy.

## 1. Introduction

Essential oils are volatile fractions of plant materials composed of phytonutrients (amino acids and essential fatty acids), secondary metabolites (polyphenols, flavonoids), antioxidants, vitamins, minerals, and other bioactive compounds. They contain concentrated chemicals that form the ‘essence’ of the plant species. For this reason, it is primarily used in aromatherapy and as flavoring agents. Additionally, due to the presence of concentrated bioactive chemicals, they are regarded as potential medicinal agents [1]. They are commonly used in combination therapies to overcome the side effects of strong medicines (as in case of chemotherapy) [2]. For instance, Mentha spicata and Mentha × piperita essential oils were found to be effective in overcoming chemotherapy-induced nausea and vomiting (CINV) in patients with different types of cancer [3]. After supplementation with Mentha spicata and Mentha x piperita essential oils, a significant reduction (*p* < 0.05) in the intensity and frequency of emetic events without any side effects in the first 24 h after chemotherapy compared to the control group and a reduction in treatment expenditure in cancer patients undergoing chemotherapy were recorded [3]. Almost every ancient culture and complementary medicinal systems including Ayurveda, Chinese, Korean as well as European origins have described the use of essential oils for over 5000 years [1,4].

Essential oils can be extracted from medicinal plants by a simple distillation process with the use of steam or cold press. For extraction of essential oils that are sensitive to heat, CO_2_ extraction is used as more a sophisticated technique to get higher yields in less time [5]. Over the years, several plants with significantly higher concentrations of essential oils have been used for medicinal purposes. However, a few well-known medicinal plants contain low concentrations of essential oils and hence, they are not considered as practical for commercial production in spite of their potential medicinal value. One such plant is known as Taraxacum officinale (TO) [6]. It is a perennial herb that belongs to the Asteraceae family and is commonly referred to as ‘dandelion’. Dandelion is a common weed that proliferates immensely in gardens, pastures, wasteland, and forests. In spite of the medicinal properties of this plant, extensive economic damage has been reported due to its weedy nature [7]. Hence, it is possible to overcome the commercial constrains of plant availability for essential oil extraction from this plant. At present, not many studies have analyzed the phytochemical constituents of TO. Among the very few studies reported in literature, gas chromatography–mass spectrometry analysis of TO essential oil obtained by hydrodistillation of flower indicated presence of **25** compounds, which were dominated by 1, 3-dimethylbenzene, 1, 2-dimethylbenzene, 1-ethyl-3-methylbenzene, heneicosane, and tricosane [6].

The Asteraceae family is well known for their antioxidant and antiproliferative properties, which are demonstrated using both in vitro and in vivo models in literature [8,9,10,11,12,13,14]. Globally, TO is one of the most widespread members of this family. The term ‘Taraxacum’ is derived from Greek vocabulary ‘taraxos’ and ‘akos’ which translate to ‘disorder’ and ‘remedy’, respectively [15]. As the name suggests, all members of this genus, consisting of over 2500 species, are botanically recognized for their medicinal properties [15]. The plant is native to Europe and distributed widely in the warm temperate zones of Northern hemisphere. It is also found commonly in regions of Asia and North America. The plant has been utilized in the treatment against various ailments such as cholera, cancer, rheumatism, scurvy, acidosis, headache, collagen buildup, jaundice, and uric acid disorders [16]. The roots and young plants are mainly used for medicinal purposes, showing potent activity against liver damage [17]. The biological properties of TO are attributed to the phytochemicals concentrated in the flowers, leaves, roots, and stem, with every part possessing biological activity against specific disorders [18,19]. The plant is a rich source of sesquiterpene lactones, triterpene, sterols, phenolic acids, and coumarins which are well known for their anticancer properties [20,21,22,23,24,25,26,27]. The phytochemicals in TO stimulate multiple cell signaling pathways by modulating the activities of various cancer-related factors like NFκB, Akt, MEK, ERK, sVCAM-1, MAPK, MMP, TNF, and IL [18,19,21,22,23,24,25,26,27].

Considering the above factors, the present study aims to determine the chemical composition of TO essential oils and evaluate its antioxidant and antiproliferative potential using both in vivo and in vitro models.

## 2. Results

### 2.1. Chemical Analysis

The hydrodistillation extraction of the *TO* provided EO with a light-yellow coloration and a weak odor. The average yield of the oil obtained was (0.071 ± 0.003% (*w*/*w*)).

The analysis of the chemical composition of the EO extracted from the *TO* identified **34** compounds elected between 2.07 and 35.989 min. Based on the intensity of the peaks, it is concluded that among the **34** compounds identified by GC-MS in the EO, **7** compounds compose the majority: **n**-hexadecanoic acid, 9,12-octadecadienoic acid, octadecadienoic acid, linoelaidic acid, pentadecanoic acid, n-nonadecanol-1, and tetradecanoic acid (Table 1).

### 2.2. Total Phenolic Compounds

Based on the measured absorbance value of the EO reacting with Folin–Ciocalteu reagent and comparison to the absorbance values for calibration standards of gallic acid (Y = 0.0133x − 0.01; R² = 0.9902), the amount of total phenolics in EO was calculated to be 4.31 ± 1.68 mg GAE/g extract.

### 2.3. Antioxidant Activity

#### 2.3.1. DPPH Assay

In order to determine the antioxidant power of *TO* EO, a DPPH free radical scavenging assay was used and scavenging activity was determined on the basis of their percentage of inhibition of the DPPH and their IC_50_ values. The mean percentage of DPPH free radical scavenging activity at different concentrations of EO are represented in the following histogram (Figure 1).

From the results obtained, it can be observed that the radical scavenging activities of the EO and the standard BHT are directly proportional to their concentration. The EO demonstrated a moderate percentage of inhibition of 25 ± 0.64% at a dose of 1 mg/mL compared to the standard antioxidant BHT, which showed a significant DPPH inhibitory power of 94.72 ± 0.03% at 800 μg/mL. The IC_50_ values found for the DPPH assay corroborate these findings. *TO* EO has a high IC_50_ value (2.00 ± 0.05 mg/mL) compared to the standard BHT (0.17 ± 0.01 mg/mL). The higher the IC_50_ value, the lower the radical scavenging activity of antioxidant potential. When compared to the synthetic standard BHT, the EO had a moderated antioxidant capacity.

A Pearson’s correlation was established between phenolic content and DPPH radical scavenging activity of the *TO* EO (Table 2).

Table 2 reveals a strong positive correlation between phenolic compound content (TPC) and DPPH radical scavenging activity (IC_50_) with Pearson’s correlation coefficients of r = 0.9669.

#### 2.3.2. Reducing Power Assay

In this experiment, the reducing power capacity of the *TO* EO and the standard AA was shown to be concentration dependent. An increase in absorbance suggests an increase in the reducing power ability (Table 3). In terms of EC_50_ value, *TO* EO showed a good reducing power potential (EC_50_ = 0.963 ± 0.006 mg/mL) but significantly (*p* < 0.05) lower than the AA standard (EC_50_ = 0.034 ± 0.28 mg/mL) (Table 3). According to the data provided here, the significant reducing power of *TO* EO appears to be due to its antioxidant capacity.

### 2.4. In Vivo Antioxidant Activity

#### 2.4.1. Effects of Treatments on Body Weight of Treated Mice

The body weights of the mice were measured at the beginning (day 0) and at the end of the test (day 14) and the findings are shown in Table 4. For the control groups (C and vehicle CMC), standard AA, and the EO (600 mg/kg and 1200 mg/kg, respectively) groups co-treated with APAP, the body weights recorded slight and normal increases throughout the treatment period, with no statistical difference between day 0 and day 14 (*p* > 0.05). However, compared to day 0, the APAP group showed a significant drop in body weight at the end of the treatment (day 14) (*p* < 0.05). It is apparent that EO has an ameliorating effect on the body weight of the mice, as shown by the slight weight gain (*p* > 0.05) in the batches treated with the combination (EO/APAP). This can be explained by the interesting nutritional value of our EO which is rich in primary metabolites.

The occurrence of body weight variations between treatment and control groups, as seen in Table 4, complicates the interpretation of organ weights. To detect target organ damage, relative organ weight to body weight was used (Table 5). Intact liver and kidney weights were converted to relative weights of 100 g of body weight.

Based on the results in Table 5, a statistically significant decrease in organ weights relative to body weight was observed in the kidneys and liver of mice subjected to APAP treatment (AA, APAP alone, *TO* 1, and *TO* 2) compared to the control group (**** *p* < 0.0001). In addition, a significant increase in organ weights relative to body weight was recorded in the kidneys and liver of APAP-exposed mice (AA, *TO* 1 and *TO* 2) compared to the APAP toxic group (### *p* < 0.001). These changes in organ weights relative to the body weight indicate a toxic effect of APAP on the animals’ organs (liver and kidneys). Supplementation with AA and *TO* EO showed a significant ability to counteract the harmful effect of APAP ### *p* < 0.001). Interestingly, a higher dose of the *TO* EO (i.e., 1200 mg/kg PC) showed a protective effect comparable to that of the AA positive control in APAP-treated mouse models (### *p* < 0.001). The above observations indicate that the hepato/renal protective activity of the studied essential oils act in a dose-dependent manner.

#### 2.4.2. Oxidant Stress Parameters Analysis

To assess the effect of oral administration of *TO* EO on the biomarkers (enzymes) of oxidative stress in the renal and hepatic tissues of mice models, we quantified the enzymatic activities of catalase (CAT), superoxide dismutase (SOD), and glutathione (GSH) and the levels of malondialdehyde (MDA) in the tissue homogenates.

Our findings show that the toxic control group (APAP) has a statistically significant increase in hepatic and renal MDA levels, a lipid peroxidation index reflecting the degree of oxidative stress, and significantly lower levels of SOD, CAT, and GSH compared to the control, standard AA, and EO-treated groups (600 and 1200 mg/kg body weight) (*p* < 0.05). The increase in MDA could be attributed to the increased generation of reactive oxygen species following APAP administration (Figure 2).

The batches treated with *TO* EO (600 and 1200 mg/kg P.C., respectively) had levels of SOD, GSH, and CAT comparable to the control (C), vehicle (CMC), and standard AA groups (Figure 2). After administration of *TO* EO, a correction of the antioxidant defense deficit marked by an increase in antioxidant enzymes (CAT, SOD and GSH) at the hepatic and renal level was observed. The APAP-treated group’s kidney homogenate showed a statistically insignificant decrease in GSH level.

Treatment with *TO* EO preserved the levels of SOD, CAT, and GSH to values comparable to the normal control (*p* > 0.05) and exerted a protective effect by reducing oxidative damage marked by a significant dose-dependent decrease in the level of MDA in liver and kidney tissue compared to the toxic APAP control (*p* < 0.05). In light of the findings, the investigated EO could be beneficial in the treatment of disorders caused by oxidative damage.

### 2.5. Cytotoxic Activity

Human cervical cancer HeLa cells were used to test the cytotoxic effect of different concentrations of *TO* EO (2–95 µg/mL). The MTT assay was used to measure the degree of cytotoxicity of the tested oil towards HeLa cells (Figure 3). Cytotoxic activity was expressed as percentage inhibition of cell viability compared with the untreated control (100%). The anticancer agent doxorubicin (DOX) was used as a standard (Figure 3).

The results, presented in Figure 3, show that the studied EO significantly (*p* < 0.001) and dose-dependently inhibited the growth of HeLa cells compared to untreated cells (100%) (Figure 3).

At the highest applied dose (95 µg/mL), a significant (*p* < 0.0001) maximum inhibitory effect of *TO* EO on HeLa cell line growth was recorded (% I: 83.58%; IC_50_ = 45.56 ± 0.05 µg/mL) but lower than that recorded by the standard DOX (% I = 92.50%; IC_50_ = 4.34 ± 0.02 µg/mL) (Figure 3). At lower doses (2–10 µg/mL), the oil was still toxic to HeLa cells with comparable inhibition to DOX.

Our essential oil exerted a remarkable antiproliferative effect on the HeLa cell line. Based on these results, it is suggested that the *TO* essential oil may be useful in the prevention and reduction of cancer occurrence and offer hope for antitumor therapeutic research.

## 3. Discussion

Essential oils are synthesized throughout the plant, either in specialized secretory channels (present in intercellular spaces) or oil glands located on the outer side of the plants [4]. In rare instances, the highest concentration of essential oil extracted from plants is reported to be 15% [28]. However, generally, the EO content of a plant is approximately 1% or lower [29]. In our study, the yield of *TO* EO was (0.071 ± 0.003% *m*/*m*). The synthesis of volatile and fragrant hydrocarbons takes place in the stem and leaves of plants, from where they migrate towards the flowers. This is because one of the functions of EOs is to attract pollinators. The EOs then accumulate in the fruits and seeds of the plant [4]. A number of studies have reported the efficacy of solvent extracts of test *TO* plant used in this study. However, very few studies have reported on the yield and composition of EOs obtained from this plant. A comparable yield of 0.08% *v*/*w* was recorded by *TO* EO from fresh flowers [6]. The difference in yield of EOs in reported literature is due to the fact that the concentration and constituents of the EO are influenced by multiple parameters. These include the soil texture, soil nutrient availability, seasonal variations, geographic locations, climatic conditions, and post-harvest storage and treatment conditions. Moreover, they also differ in composition and concentration at different stages of plant and fruit development [30].

The GC-MS analysis of *TO* EO revealed the presence of 34 components in the current study. Among these compounds, 8 major constituents were identified. Diterpene monocyclic alcohols and fatty acids made up the majority of the chemicals found. The bioactivities of the major constituents identified in the oil of test plant material are represented in Table 6 below.

The estimation of the total phenols in plant extracts gives a significant impression of their biological activity as well as the suitability in preparations of various medicinal or non-medicinal mixtures. In addition to medicinal properties, the plants with higher phenolic content are more likely to be resistant to changes in temperature, light intensity, pathogenic infections, etc.s [41]. In the present study, the *TO* EO showed a low concentration of phenol (4.31 ± 1.68 mg GAE/g extract). While the total phenolic content of solvent extracts is commonly reported in literature, no study has documented the total phenolic content of the *TO* EO. The ethanolic extract of dandelion root from Chirpan, South-Central Bulgaria, using the Soxhlet extraction method, showed a comparable TPC value of 4.5 ± 0.1 mg GAE/g extract to our findings [42]. Conversely, other research has reported a higher total phenolic content of methanolic extract from Bucheon, Korea, and ethylic extract of *TO* from Erbil, northern Iraq, with TPC values of 176.8 and 10 mg GAE /g dw, respectively, using a large-scale extractor and maceration process, respectively [43,44]. A lower TPC value than ours, ranging from 0.229 ± 0.010 to 0.535 ± 0.033 mg CE/g DW was recorded in the *TO* leaves and flowers (acetone and triton) extracts from Rzeszów, Poland, using the micelle-mediated extraction method [45]. This difference in the total phenolic values could arise from the genetic and/or growing location, climate, maturity, and harvest season variation [46].

In the present study, the antioxidant activity of essential oil was tested by the DPPH radical scavenging method using BHT as the positive control. The dose-dependent antioxidant activity observed suggests that the DPPH radical scavenging activity is directly proportional with the concentration of essential oil and reference standards. Based on our observations of radical scavenging activity and IC_50_ values, it was concluded that *TO* possesses weak antioxidant activity. The values obtained were not significant compared to the reference standards. Similar to our findings, a weak antioxidation potential of *TO* has been reported by Tettey et al. Hexanic and methylene chloride *TO* leaf extracts showed a DPPH scavenging of 22.7 ± 0.4% and 23.3 ± 0.4%, respectively [47]. In the same manner, the *TO* fruit ethanolic extract revealed an inhibition of 37.7% of DPPH at a concentration of 75 µg/mL [48]. Unlike our study, strong antioxidant potential of *TO* has been also reported in the literature [49]. Paduret et al. found a 80.664% capacity of inhibition of DPPH solution for *TO* methanolic extract [50]. At a concentration of 100 g/mL, the ethanolic extract of *TO* flowers recorded an inhibition of 90.27 ± 0.5% [51]. Aqueous extract showed a lower IC_50_ of 4.48 µg/mL than our finding [52]. The weak activity of the essential oil may be due to comparatively lower concentration of phenols [53]. The statistical findings support this hypothesis; in our investigation, a strong positive correlation was found between the phenol content of essential oil of TO and its antioxidant activity.

In contrast to the DPPH assay, the *TO* oil showed a modest reducing power with an EC_50_ of 0.963 mg/mL. A higher reducing power was reported elsewhere for the *TO* root extract at an IC_50_ of 0.138 ± 0.001 mg/mL [54]. Lower reducing power of essential oils for other plants within the *Asteraceae* family was reported in the literature. The reducing power of *Silybum marianum* seeds oil showed a comparable EC50 to our study of 1 mg/mL [55]. Conversely, the *Artemisia herba-alba* essential oil exhibited a weak reducing power with an EC_50_ ranging from 1.2 to 2.9 mg/mL [56]. Recently, Dhouibi et al., 2020 reported a lower reducing power than ours of EC_50_ > 20 mg/mL for the essential oil of two *Centaurea* species, *C. kroumirensis Coss.*, and *C. sicula* L. subsp *sicula* [57]. The *TO* essential oil’s reducing power in our investigation appears to be linked to the presence of antioxidant components. Hexadecanoic acid and 9,12-octadecadienoic acid found in the studied oil have been previously described for their antioxidant properties [35].

The ingestion of *TO* EO was accompanied by a slight weight gain in the mice. Although the weight gain was not significant, the short time span of the trial distinctly highlights the nutritional value of the EO. This may suggest that the EO exerted a protective effect on liver and kidney, more pronounced for the higher dose (i.e., 1200 mg/kg PC). This would indicate that the nutritive and hepato/renal protective activity of essential oils is dose dependent.

Regarding the study of oxidative stress biomarkers in the renal and hepatic tissues, we found decreased SOD, CAT, and GSH levels and increased MDA levels in the APAP group. The resulting oxidative stress was mitigated in the mice groups treated with *TO* EO with enzyme levels comparable to those of standard controls. The co-administration of *TO* oil in mice groups with APAP-induced oxidative stress significantly decreased the MDA level and restored the antioxidant enzymes (CAT, SOD, and GSH) to levels that were comparable to controls.

Antioxidant potential is a major part of the innate immune system in humans and animals [58,59]. The reactive oxygen species generated in different tissues as a result of metabolic activity are counteracted by the antioxidant enzymes and neutralized [60]. Each antioxidant enzyme specifically neutralizes a type of ROS contributing to the overall activity of the immune system [61]. SOD, CAT, and GSH act on superoxide, peroxide, and singlet oxygen radicals, respectively, to convert them into molecular oxygen and/or water [62]. In the absence of an effective antioxidant system, the ROSs cause the peroxidation of the membrane polyunsaturated fatty acids [63]. The kidney and liver are the main centers of metabolic activity, and hence the changes in enzyme activity are more apparent in these tissues. The increase in the levels of antioxidant enzymes after administration of EO in our study clearly demonstrates its role in the inhibition of ROS. As observed in the APAP group, an eventual inability of the EO to overcome the oxidative stress would have resulted in a similar triggering of the inflammatory response in the test groups, atrophy of hepatic and renal tissues, and significant reduction in the level of antioxidant enzymes. However, despite the moderate antioxidant activity observed in the in vitro experimental conditions, the *TO* EO demonstrated overall a significant antioxidant activity in the APAP-induced oxidative stress mice model.

SOD is primarily distributed in the cytosol, but it is also detected in lysosomes, peroxisomes, mitochondria, and the nucleus [64]. Several subtypes of SOD also exist, and a high degree of specificity for each subtype of SOD is recorded in the extracellular matrix of specialized tissues including that of lung, heart, kidney, plasma, lymph, ascites, and cerebrospinal fluid [65]. Interestingly, a specific type of SOD (known as MnSOD) is reported to decrease in cancer cells, suggesting its critical role in cancer prevention [66]. Moreover, studies have reported reversal of cancer pathology to normal cell physiology upon administration of MnSOD in vivo models [67]. Similar anticancer effects are also reported for CAT and GSH [68,69]. The CAT enzyme is largely located in the subcellular organelles known as peroxisomes [70]. CAT has been found to have anti-tumor properties [71,72]. It catalyzes the dismutation of hydrogen peroxide, which is a cell growth stimulator and a secondary messenger of mitogenic signaling cascades [73]. It also inhibits cell proliferation and the activation of growth factor-dependent mitogen-activated protein kinases ‘MAPKs’ [74]. Gal-catalase (galactosylated) and Suc-catalase (succinylated), catalase derivatives, have been discovered to reduce liver surface metastasis by suppressing nuclear factor B (NF-B) activity in liver and tumor cells [75].

Another marker of oxidative stress is MDA, which indicates lipid peroxidation of membrane structure and cellular injury due to depletion of the endogenous antioxidant enzymes [17]. Patients with breast, laryngeal, lung, and oral cancers have been reported to have high levels of MDA [76,77]. This byproduct has been described as a co-carcinogen and tumor promoter due to its high reactivity and cytotoxicity, and as a result, destabilizes the membrane structure and formation of carcinogen–DNA adducts due to its reaction with cellular components including DNA [75,78,79].

The in vivo antioxidant results of this article corroborate with the findings of Colle et al. on ethanolic *TO* extract administration to mice preventing liver tissue damage and alterations in biochemical parameters caused by APAP [48]. Colle et al. attributed the antioxidant activity of *TO* to the electron transfer ability of its phenolic compounds [48]. In our study, the capacity of *TO* oil to thwart the APAP-induced oxidative stress in the liver and kidney could be attributed to its chemical composition, which has previously documented for its potent antioxidant activity (Table 6). The hepatoprotective effects of plant essential oils are frequently reported on in the literature due to their antioxidant properties. Essential oils extracted from fennel, cumin, and flower buds of clove the level of antioxidant enzymes in cyclophosphamide-induced hepatotoxic mice models were shown to restore the function of liver cells [80]. The mechanism of antioxidant enzyme induction was described in a study of Zou et al. on the efficacy of oregano essential oil to overcome the deleterious cellular effects of the specifical induction of SOD1 and GSH occurring following the activation of the nuclear factor-erythroid 2-related factor-2 after oil exposure in peroxide-induced oxidative damage models of porcine small intestinal epithelial cells [81].

The cytotoxic activities of *TO* EO were determined by MTT assay using HeLa (cervical cancer) cell lines. The antiproliferative effects of the oil were significantly visible in the HeLa cell lines. *TO* oil demonstrated the best antiproliferative activity at a concentration of 95 µg/mL, with an inhibition percentage of 83.53% and an IC_50_ of 45.56 ± 0.05 µg/mL. Since MTT reduction indicates mitochondrial/non-mitochondrial dehydrogenase activity, in showing a positive MTT assay, the essential oil (as an indication of cytotoxic activity) also indicates a potential to overwhelm the enzymatic activity of mitochondria. This insight is indicative of substantial injury to the cancer cells and the efficacy of test samples [82]. Previous research has shown that *TO* possesses anticancer properties, which supports our findings. Aqueous dandelion root extract was shown to have anticancer activity in numerous carcinoma cell types in vitro, with no damage to non-cancer cells [83]. At a concentration of 50 g/mL, the aqueous *TO* root extract inhibited and reduced the cell viability of MCF-7/AZ breast cancer cells by 50% via inhibition of the dual kinase complex focal adhesion kinase (FAK)-steroid receptor coactivator (Src) and downregulation of matrix metalloproteinases MMP-2 and MMP-9 [84]. In human hepatocellular carcinoma cells, the methanolic extract of *TO* demonstrated significant anti-carcinoma activity [85]. Huh7 cell viability was reduced significantly when *TO* and the TNF-related apoptosis-inducing ligand ‘TRAIL’ (a cytokine that promotes apoptosis in cancer cells) were combined when compared to TRAIL treatment alone, with no effect on the viability of cells [85]. This suggests that *TO* may act as a novel TRAIL, sensitizing carcinoma cells to TRAIL-induced apoptosis by blocking mitogen protein kinase kinase 7-TOR signaling pathway regulator-like (MKK7-TIPRL) interaction and activating JNK [85]. The root extracts of *TO* induce apoptosis and autophagy in BxPC-3 and PANC-1 pancreatic cancer cells with no effect on non-cancerous cells [86]. Furthermore, Hexadecanoic acid, a key component of our EO, has been shown to induce apoptosis in HT–29 colon cancer cells [31].

In general, the cytotoxic effects of essential oils are mediated through the induction of cell death [87]. This mechanism is activated by sequential activation of molecules that trigger apoptosis/necrosis, cell cycle arrest, and loss of function of essential organelles [88]. However, the most recognized benefit of essential oils in exhibiting cytotoxic effects can be attributed to its lipophilic nature and biochemical makeup of low-molecular-weight components [89]. These properties of essential oil readily allow the alteration in membrane composition, which in turn, increases the fluidity of the cell membrane and allows the individual bioactive molecules of essential oils to enter the cell. Consequently, this results in the leakage of ions and cytoplasmic components. Moreover, the essential oil also causes alterations in the pH gradient and loss of mitochondrial/cellular redox potential that compromise ATP production in cell membranes. Ultimately, this results in cell death [88,90].

Existing literature has highlighted the mechanism and activity of very few essential oils as anti-proliferative. The fact that the essential oil used in our study exhibits both antiproliferative and antioxidant activities indicates a high possibility of its use in cancer treatment. Together, these two properties may also enhance cellular immunity by activation of detoxification systems and DNA repair [2]. Several studies have also identified diverse pathways of anti-proliferative mechanisms of key oil components in cancer cell lines [91,92,93,94]. The most common mechanism is the induction of apoptosis leading to cytoskeletal alterations, plasma membrane damage, mitochondrial dysfunction, DNA fragmentation, caspase-3 activation, and cleavage of pro-survival proteins [91,92,93,94]. Although these mechanisms are not entirely understood, they seem to be effective against glioblastoma, melanoma, leukemia, bone, breast, lung, ovary, pancreas, and prostate cancers [95]. The anti-proliferative and apoptotic effects of the thunbergol diterpene present in *TO* EO was confirmed in J82 (Blc) cell lines and human melanoma and renal UO-31 cancer cells [39,96]. Additionally, the plant used in our studies is considered to be an adjuvant drug in traditional therapies to enhance immunity [97]. In addition, their anti-tumor and anticancer activities are also extensively reported [84,97,98]. These factors collectively favor the potential use of essential oils in cancer therapies.

## 4. Material and Methods

### 4.1. Plant Material

*TO* samples were collected from El-Jadida, Morocco. For future reference, a specimen was deposited at the Faculty of Sciences and Technologies, Hassan 1st University, Settat, Morocco (Voucher n° 0358/M). The plant was carefully cleaned with sterile distilled water to remove dust and foreign matter, then shade dried and processed to a fine powder using an electric grinder.

### 4.2. Isolation of Essential Oil

The hydrodistillation was performed as previously demonstrated by our group [99].

### 4.3. Chemicals and Reagents

Doxorubicin, superoxide dismutase (SOD), iron chloride (FeCl_3_), nitro blue tetrazolium (NBT), ascorbic acid (AA), catalase (CAT) from bovine liver, acetaminophen (APAP), 2,2′-Diphenyl-1-picrylhydrazyl (DPPH), **n**-alkanes (C6–C30), thiobarbituric acid (TBA), trichloroacetic acid (TCA), ethylenediaminetetra acetic acid (EDTA), nicotinamide-adenine dinucleotide phosphate (NADPH), butylated hydroxytoluene, gallic acid, sodium carboxymethyl cellulose (CMC), phenazine methosulfate, Folin–Ciocalteu reagent, reduced glutathione (GSH), and 1,2-dithio-bis nitro benzoic acid (DTNB) were purchased from Sigma Co. (St. Louis, MO, USA). Dimethyl sulfoxide (DMSO) was purchased from Merck (Merck, Darmstadt, Germany). Streptomycin, Dulbecco’s modified Eagle’s medium (DMEM), and streptomycin were purchased from Biochrom (Biochrome AG, Berlin, Germany, A 321-44). Fetal bovine serum was purchased from Sigma (Sigma, Darmstadt, Germany). Potassium phosphate monobasic, sodium pyrophosphate dibasic, sodium carbonate (Na_2_CO_3_), disodium hydrogen phosphate (Na_2_HPO_4_), hydrogen peroxide (H_2_O_2_), potassium ferricyanide (K_3_Fe(CN)_6_), sodium sulphate anhydrous (Na_2_SO_4_), acetic acid (ACA), and n-butanol (99.8%) were of analytical grade and purchased from Merck (Nottingham, UK).

### 4.4. Gas Chromatography–Mass Spectrometry of Essential Oil

The GC-MS was conducted as previously described by our group [99].

### 4.5. Determination of Total Phenolic Content

The total phenolic was evaluated as previously described by our group [99].

### 4.6. In Vitro Antioxidant Activity

#### DPPH and Reducing Power (RP) Assays

The 2,2-diphenyl-1-picrylhydrazyl (DPPH) and reducing power (RP) assays were conducted as previously described by our group [99].

### 4.7. In Vivo Antioxidant Activity

#### 4.7.1. Animal Models and Induction of Oxidative Stress

Animal models and the induction of oxidative stress were previously described by our group [99].

#### 4.7.2. Experimental Design

The experimental design was evaluated as previously mentioned by our group [99]. For this purpose, mice were randomly divided into 5 groups (*n* = 5):Group I was designated as vehicle and was treated with 0.1% CMC;Group II (negative control) received no treatment but had free access to water and food;Groups III (toxic control), IV, V, and VI received a single intraperitoneal injection of acetaminophen (APAP) (400 mg/kg, ip) before the start of the experiment to induce hepato-renal oxidative injury;Group IV served as the standard and received AA, 200 mg/kg body weight;Groups V and VI received *TO* EO at doses of 600 and 1200 mg/kg body weight.

These doses were chosen following a screening procedure in which we tested three doses (600, 1200, and 1600 mg/kg) of the EO by oral administration to mice for 2 weeks (unpublished results). At the highest concentration (1600 mg/kg), we observed the installation of the LD_50_, whereas the other two doses did not induce signs or symptoms of toxicity and were selected for further antioxidant studies. Animals were treated orally once daily (at 9 am) for two weeks (14 days). A 10-day quarantine was observed before treatment [100].

The experiment was conducted in accordance with the Guide for the Care and Use of Laboratory Animals and approved by the Bioethics Advisory Commission of the Faculty of Sciences of Agadir (CCBE-FSA Ref. Ref. No: Ref. No: ER-BS-04/2022-0001).

#### 4.7.3. Preparation of Tissue Homogenates

The preparation of tissue homogenates was performed as previously described by our group [99].

#### 4.7.4. Body and Organ Weights

The body and organ weights were evaluated following the protocol previously described by our group [99].

#### 4.7.5. Quantification of Oxidative Stress Biomarkers in Tissue Homogenates

The enzymatic activities of catalase, superoxide dismutase (SOD), glutathione GSH, and malondialdehyde MDA were assessed following the protocol previously described by our group [99].

### 4.8. Cytotoxic Assay

#### 4.8.1. Sample Preparation

The essential oil was resuspended in 0.1% DMSO at a final concentration of (1.0 mg/mL). The positive control doxorubicin (DOX) 1MG (44583, Sigma-Aldrich) was prepared as stock solutions (1 mg/mL) in Nacl 0.9%. The working solutions of the oil and DOX were prepared by further dilution of the stock with the Dulbecco’s modified Eagle’s medium to concentrations of 2, 5, 10, 25, 45, 55, 75, 85, and 95 µg/mL. The samples were filtered using 0.22 µm Syringe Filter, PVDF (Sterile) (Starlab Scientific, Cape Town, South Africa).

#### 4.8.2. Cellular Assays

##### Cell Lines

The human cervical epithelioid carcinoma HeLa (CCL-2, ATCC) cells were cultured in Dulbecco’s modified Eagle’s medium (DMEM, T043-10, Biochrom AG), supplemented with fetal bovine serum 10 % (*v*/*v*) (FBS, Sigma, Darmstadt Germany, F9665), penicillin 100 IU/mL (Biochrom AG, Berlin, Germany, A 331-26) and streptomycin 100 μg/mL (Biochrom AG, Berlin, Germany, A 321-44). The cell cultures were seeded at 10.000 cells/cm^2^ and maintained in a CO_2_ incubator (Nuve EC 160, Ankara, Turkey) at 37 °C in 5% CO_2_. Cell viability was determined using the trypan blue dye exclusion method.

#### 4.8.3. Cytotoxic Assay

The MTT (3-(4,5-dimethylthiazol-2-yl)-2,5-diphenyltetrazolium bromide) assay was used to evaluate the cytotoxic effects of *TO* EO against Hela cell line [101]. The cell cultures (cells/mL) were plated in 96-well flat-bottomed plates (92096, TPP, Switzerland) overnight. After the medium was discarded and cells were washed, the renewal complete medium was supplemented with different concentrations of the tested oil and the positive control doxorubicin (DOX) in doses of 2, 5, 10, 25, 45, 55, 75, 85, and 95 µg/mL and incubated for 24 h (the well containing the complete medium with 0.1 % DMSO was used as a control group). After treatment time expired, the medium was discarded, cells were washed with PBS (phosphate-buffered saline), and 100 µL of fresh medium containing 10 µL of MTT (5 mg/mL) was added in each well and incubated for 3 h. To solubilize the formed formazan crystals, 100 µL DMSO (dimethyl sulfoxide, 1029521000, Merck, Darmstadt, Germany) was added and the absorbance was read at 540 nm (after 10 min of shaking at 250 rpm) by an ELISA reader microplate (EZ400 reader, Biochrom, Cambridge, UK). All experiments were performed in triplicate (*n* = 5). Cellular viability (%) and inhibition (%) were calculated according to the formula:% cell viability = (Absorbance_sample_/Absorbance_control_) × 100.
%I = 100 [1 − (Absorbance_sample_/Absorbance_control_)]

The IC_50_ value of the essential oil was determined by curve fitting and was used as a criterion to judge cytotoxicity.

### 4.9. Statistical Analysis

The experiments were replicated three times. The data were statistically analyzed using one way analysis of variance (ANOVA) followed by Student’s *t*-test analysis using GraphPad Prism 8.4.3 (GraphPad Software Inc., San Diego, CA, USA), and the values were expressed as mean ± SD. Linear regression analysis was used to calculate the IC_50_ values. Pearson’s correlation coefficient was calculated using GraphPad Prism 8.4.3 (GraphPad Software Inc., San Diego, CA, USA). Values of *p* < 0.0001 were considered statistically significant.

## Figures and Tables

**Figure 1 molecules-27-06477-f001:**
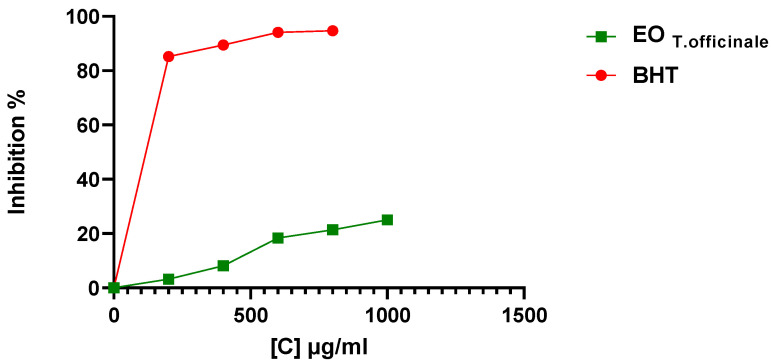
Percentage inhibition of DPPH by different concentrations of *Taraxacum officinale* essential oil and the standard BHT. Results are expressed as means ± SD of three parallel measurements *p* < 0.01. BHT: butylated hydroxytoluene; EO: essential oil.

**Figure 2 molecules-27-06477-f002:**
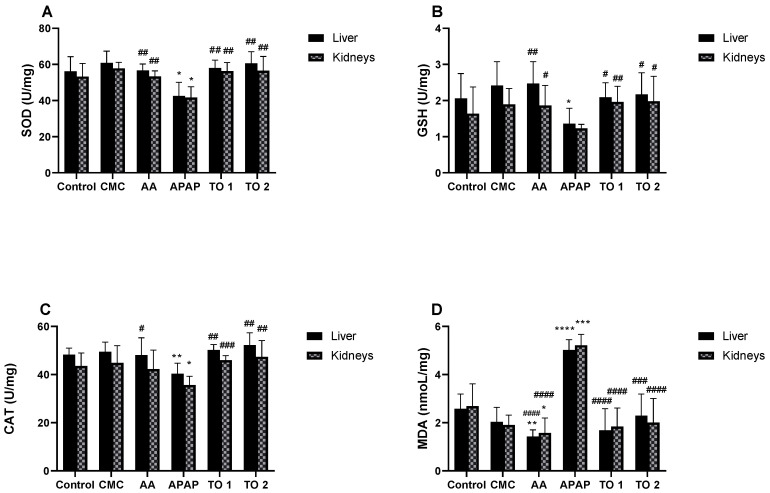
Effect of *Taraxacum officinale* essential oil on antioxidant enzymes and MDA levels against acetaminophen-induced liver injury in mice. Values are expressed as means ± SD (*n* = 5), * *p* < 0.05; ** *p* < 0.01; *** *p* < 0.001; **** *p* < 0.0001 vs. the normal control group and # *p* < 0.05; ## *p* < 0.01; ### *p* < 0.001; #### *p* < 0.0001 vs. the toxic control group. (**A**) Effect of *Taraxacum officinale* EO on SOD level in liver and kidney of APAP-treated mice; (**B**) effect of *Taraxacum officinale* EO on GSH level in liver and kidney of APAP-treated mice; (**C**) effect of *Taraxacum officinale* EO on CAT level in liver and kidney of APAP-treated mice; (**D**) effect of *Taraxacum officinale* EO on MDA level in liver and kidney of APAP-treated mice. SOD: superoxide dismutase; GSH: reduced glutathione; CAT: catalase; MDA: malondialdehyde; AA: ascorbic acid; CMC: sodium carboxymethylcellulose; APAP: acetaminophen; *TO* 1: *Taraxacum officinale* essential oil 600 mg/kg body weight; *TO* 2: *Taraxacum officinale* essential oil 1200 mg/kg body weight.

**Figure 3 molecules-27-06477-f003:**
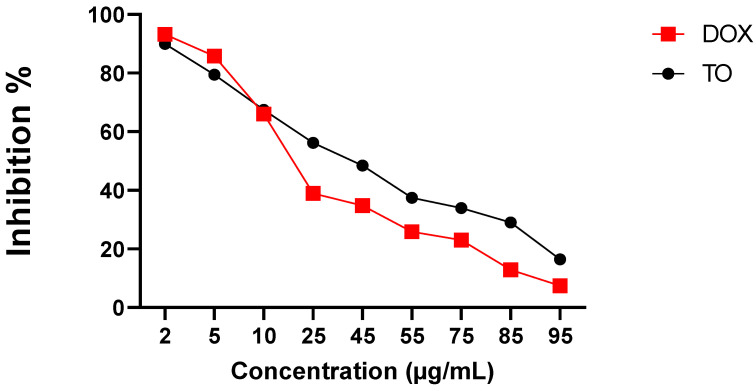
Effect of *Taraxacum officinale* EO on the growth of HeLa cell line. TO: *Taraxacum officinale* essential oil; DOX: standard doxorubicin (results are expressed as mean ± SD. (*n* = 5).

**Table 1 molecules-27-06477-t001:** Chemical composition of *Taraxacum officinale* essential oil.

Name	^a^ RI	^b^ RI	% Area
Pentadecanoic acid	1762	1777	2.28
Tetradecanoic acid	1774	1774	0.99
**n**-Hexadecanoic acid	1987	1980	26.11
Thunbergol	2051	2047	0.66
Heptadecanoic acid	2081	2080	0.81
Heptadecanolide	2094	2051	0.95
9,12-Octadecadienoic acid	2105	2152	34.19
n-Nonadecanol-1	2157	2153	1.36
Octadecanoic acid	2205	2165	1.11
Linoelaidic acid	2206	-	2.57

^a^ RI: retention index measured relative to **n**-alkanes (C6–C30) on the non-polar 123 DB11 column. ^b^ Linear retention index taken from the NIST 05 library.

**Table 2 molecules-27-06477-t002:** Pearson’s correlation coefficients between phenolic content and DPPH radical scavenging activity (IC_50_) of *Taraxacum officinale* essential oil.

Correlation Pearson r	Phenolic Content	DPPH
**Phenolic content**	1	0.966
**DPPH**	0.966	1

**Table 3 molecules-27-06477-t003:** Reducing ability and EC_50_ values of *Taraxacum officinale* essential oil and ascorbic acid at different concentrations.

Concentration (μg/mL)	Ascorbic Acid	EO
1000	1.52 ± 0.005	0.64 ± 0.003 *
800	1.21 ± 0.01	0.55 ± 0.04 *
600	0.95 ± 0.03	0.48 ± 0.03 *
400	0.71 ± 0.01	0.42 ± 0.05 *
200	0.43 ± 0.01	0.34 ± 0.04 *
0	0	0
EC_50_ (mg/mL)	0.034 ± 0.28	0.963 ± 0.006

Values are expressed as mean ± SD, *n* = 3, * *p* < 0.05 vs. standard. AA: ascorbic acid; EO: essential oil.

**Table 4 molecules-27-06477-t004:** Body weight of mice during the treatment period.

Treatments	Mean Body Weight in Grams ± SD	
	Day 0	Day 14
C	29.39 ± 0.29	29.58 ± 0.24
CMC	30.48 ± 0.31	30.71 ± 0.30
APAP	32.54 ± 0.43	29.78 ± 0.65 *
AA	27.47 ± 0.28	27.92 ± 0.72
*TO* 1	30.24 ± 0.22	31.8 ± 0.71
*TO* 2	23.39 ± 0.27	23.81 ± 0.25

C: normal control; CMC: vehicle–carboxymethylcellulose group 0.1%; APAP: acetaminophen-treated toxic control 400 mg/kg body weight (ip); *TO* 1: *Taraxacum officinale* essential oil 600 mg/kg body weight; *TO* 2: *Taraxacum officinale* essential oil 1200 mg/kg body weight; AA: ascorbic acid 200 mg/kg body weight. All data are means ± S.D. (*n* = 5/group), * *p* < 0.05 APAP at day 0 vs. APAP at day 14.

**Table 5 molecules-27-06477-t005:** Relative organ weight to body weight of Swiss albino mice receiving *Taraxacum officinale* essential oil for 14 days.

Groups/Organs	Relative Weight of Liver and Kidney (g/100 g)	
	Liver	Kidneys
C	5.26 ± 0.26	1.37 ± 0.11
CMC	5.05 ± 0.11	1.36 ± 0.17
APAP	3.88 ± 0.13 ***	1.04 ± 0.14 **
AA	4.66 ± 0.16 **; ###	1.24 ± 0.08 *; #
*TO* 1	4.35 ± 0.55 ****, ###	1.20 ± 0.13 *; #
*TO* 2	4.61 ± 0.07 ***; ###	1.22 ± 0.09 *; #

All values are expressed as means ± SD. C: normal control; CMC: vehicle–carboxymethylcellulose group 0.1%; APAP: acetaminophen-treated toxic control 400 mg/kg body weight (ip); *TO* 1: *Taraxacum officinale* essential oil 600 mg/kg body weight; *TO* 2: *Taraxacum officinale* essential oil 1200 mg/kg body weight; AA: ascorbic acid 200 mg/kg body weight (significant differences from normal control group * *p* < 0.05; ** *p* < 0.01; *** *p* < 0.001; and **** *p* < 0.001 significant differences from toxic control group # *p* < 0.05; ### *p* < 0.001).

**Table 6 molecules-27-06477-t006:** Bioactivities of the main constituents of the essential oil of the plant reported in the literature.

Groups	Examples of Compounds Identified in Our Study	Bioactive Potential	Reference
Fatty acid	**n**-Hexadecanoic acid; 9,12-Octa-decadienoic acid; Octadecanoic acid; Linoelaidic acid; Tetradecanoic acid; Pentadecanoic acid	Anticancer, antioxidative, immunostimulatory, anti-inflammatory, and anti-obesity	[31,32,33,34,35,36]
Fatty alcohol	**n**-Nonadecanol-1	Antioxidative and anti-obesity	[37]
Diterpene monocyclic alcohol	Thunbergol	Anticancer, antiproliferative, anti-inflammatory, and cardioprotective	[38,39,40]

## Data Availability

All data used are available on request.

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
