# Peer review of "Chemical Composition, Antioxidant and Antiproliferative Activities of Taraxacum officinale Essential Oil"

_molecules, 2022, doi:10.3390/molecules27196477_

Round 1

Reviewer 1 Report

Kamal et al. examined the composition, antioxidant and antiproliferative properties of Taraxacum officinale essential oil. Research has shown that TO essential oil can be considered in further research on anti-cancer therapy not only in terms of inhibition of cell proliferation in tumors, but also in terms of potential antioxidant protective and hepato / renal protective activity.

The authors write in the introduction that essential oils "... are commonly used in combination therapies to overcome the side effects of a strong medicine (as in case of chemotherapy)". Could the authors mention such therapeutic protocols used by clinicians to treat cancer that recommend the use of essential oils to eliminate the side effects of chemotherapy?

127 line: should be "Pearson's"

Explain abstractly the acronym APAP

Author Response

Reviewer 1

Dear Reviewer,

Thank you very much for all your notes, time, efforts, and support in improving our paper; we have carefully read the comments and have revised/ completed the manuscript accordingly. Our responses are given in a point-by-point manner below (red), as well, all the changes to the manuscript are highlighted in yellow.

Kamal et al. examined the composition, antioxidant and antiproliferative properties of Taraxacum officinale essential oil. Research has shown that TO essential oil can be considered in further research on anti-cancer therapy not only in terms of inhibition of cell proliferation in tumors, but also in terms of potential antioxidant protective and hepato / renal protective activity.

  1. The authors write in the introduction that essential oils "... are commonly used in combination therapies to overcome the side effects of a strong medicine (as in case of chemotherapy)". Could the authors mention such therapeutic protocols used by clinicians to treat cancer that recommend the use of essential oils to eliminate the side effects of chemotherapy?

Thank you for your suggestion, we have added more information to the Introduction section. Please see the yellow completions in the manuscript.

For instance, Mentha spicata and Mentha x piperita essential oils were found to be effective in overcoming chemotherapy-induced nausea and vomiting (CINV) in patients with different types of cancer. After supplementation with Mentha spicata and Mentha x piperita essential oils, a significant reduction (P < 0.05) in the intensity and frequency of emetic events without any side effects in the first 24 hours after chemotherapy compared to the control group, as well as a reduction in treatment expenditure in cancer patients undergoing chemotherapy was recorded [3].

  1. 127 line: should be "Pearson's"

Thank you very much for your comments, we have corrected it

  1. Explain abstractly the acronym APAP

Thank you for your valuable suggestion, APAP is an abbreviation referring to acetaminophen, In the abstract, we replaced the abreviation APAP with the full form « acetaminophen » as well as in the Materials and Methods section, we included the full form before the acronym APAP.

Reviewer 2 Report

This is an interesting investigation that seeks to determine the composition of volatile elements that that constitute the essential oils of the well-known "dandelion" plant.  Beyond the interesting bioactivity that is reported, some aspects related to the definition of volatile compounds that make up essential oils should be considered, and with this, consider the possibility of at least comparing the format for obtaining these metabolites with those studies that report extraction from fresh and separated material (leaves, stem, flowers).

In general, just a few inconsistencies or misleadings in the text:

ty, pathogenic infections etc [39]. In the present study, the TO EO showed a low concen-270

A comma is lacking after the word infections and before et cetera.

ue of 4.5 0.1 mg GAE/g extract to our findings [40]. Conversely, other research reported 275

The sign +- is lacking after 4.5

Tettey et al.,. The hexanic and the methylene chloride TO leaf ex-291

comma and period are left over in the sentence

aqueous extract showed a lower IC50 of 4.48 μg/mL than our finding (The lower the IC50, 298 the higher the antioxidant capacity) [50]. The weak activity of the essential oil may be 299

Sentence  (The lower the IC50, the higher the antioxidant capacity) is repeated before table 2.

Also I suggest to read and uniformize units or chemical names of compounds, considering the subscript (Examples are CO2, IC50 and other in the text, as was in the line 298.

The fact that the essential oil used in our study exhibits both 423 these activities indicates a high possibility of its use in cancer treatment. Together, these 424

I believe that for a precise reading, it should be indicated here what these two activities are (Antiproliferative and antioxidant?)

Author Response

Reviewer 2

Dear Reviewer,

We would like to thank you for classification of the manuscript as interesting. We sincerely appreciate all valuable comments and suggestions, which helped us to improve the quality of the manuscript. Appropriate changes were made and highlighted in yellow in the revised manuscript, as well, our responses are given in a point-by-point manner below.

This is an interesting investigation that seeks to determine the composition of volatile elements that that constitute the essential oils of the well-known "dandelion" plant.  Beyond the interesting bioactivity that is reported, some aspects related to the definition of volatile compounds that make up essential oils should be considered, and with this, consider the possibility of at least comparing the format for obtaining these metabolites with those studies that report extraction from fresh and separated material (leaves, stem, flowers).

  1. In general, just a few inconsistencies or misleadings in the text:

  • ty, pathogenic infections etc [39]. In the present study, the TO EO showed a low concen-270. A comma is lacking after the word infections and before et cetera.

Thank you for your kind comment. We added the missing comma

  • ue of 4.5 0.1 mg GAE/g extract to our findings [40]. Conversely, other research reported 275. The sign +- is lacking after 4.5

Thank you for your kind comment. We added the missing Symbol « ± »

  • Tettey et al.,. The hexanic and the methylene chloride TO leaf ex-291 comma and period are left over in the sentence

Thank you very much for your comment. We have removed the comma and the period

  • aqueous extract showed a lower IC50of 4.48 μg/mL than our finding (The lower the IC50, 298 the higher the antioxidant capacity) [50]. The weak activity of the essential oil may be 299 Sentence  (The lower the IC50, the higher the antioxidant capacity) is repeated before table 2.

Thank you for your suggestion, we have removed the repeated sentence

  • Also I suggest to read and uniformize units or chemical names of compounds, considering the subscript (Examples are CO2, IC50 and other in the text, as was in the line 298.

Thank you a lot for your comment. We have uniformized the units as well the chemical names of compounds

  • The fact that the essential oil used in our study exhibits both 423 these activities indicates a high possibility of its use in cancer treatment. Together, these 424. I believe that for a precise reading, it should be indicated here what these two activities are (Antiproliferative and antioxidant?)

           Thank you for your input. We indicated the activities studied.